# Mass Spectra Fitting as Diagnostic Tool for Magnetron Plasmas Generated in Ar and Ar/H_2_ Gases with Tungsten Targets

**DOI:** 10.3390/molecules28155664

**Published:** 2023-07-26

**Authors:** Cristina Craciun, Silviu Daniel Stoica, Bogdana Maria Mitu, Tomy Acsente, Gheorghe Dinescu

**Affiliations:** 1National Institute for Lasers, Plasma and Radiation Physics, 409 Atomistilor Str., 077125 Magurele, Ilfov, Romania; cristina.craciun@inflpr.ro (C.C.); daniel.stoica@inflpr.ro (S.D.S.); tomy@infim.ro (T.A.); 2Doctoral School of Physics, Faculty of Physics, University of Bucharest, 405 Atomistilor Str., 077125 Magurele, Ilfov, Romania

**Keywords:** mass spectrometry, tungsten plasma, magnetron sputtering

## Abstract

In this work, we describe an ion mass spectra processing method from plasmas generated in Ar and Ar/H_2_ gases in contact with tungsten surfaces. For this purpose, advanced model functions, i.e., those suitable for fitting the experimental mass peak profiles, are used. In addition, the peak positions, peak shapes, abundances, and ion ratios are the parameters considered for building these functions. In the case of a multielement magnetron target, the calibration of the mass spectra with respect to the peak shape and position on the *m*/*z* scale is helpful in reducing the number of free variables during fitting. The mass spectra fitting procedure is validated by the obtained isotopic abundances of W ions in W/Ar magnetron plasmas, which, in turn, are comparable with their natural abundance. Moreover, its usefulness is exemplified by calculating the ratio of WH^+^/W^+^ ions in W/Ar/H_2_ plasma. This work paves the way for obtaining relevant results regarding ion species in plasma even in the case of using general-purpose mass spectrometers with limited resolution and accuracy. Although this method is illustrated for the W/Ar/H_2_ plasma system, it can be easily extendable to any plasma type.

## 1. Introduction

Mass spectrometry is one of the major diagnostic tools in plasma physics. It is widely used to identify elements, atoms, molecules, and their ionization states, and to determine the plasma species’ molecular weight and their isotopic abundances. By injecting small amounts of solid or liquid materials such as polymers, proteins, and lipids in plasma via spraying, vaporization, ablation [1], or sputtering [2], the variety of sample types that can be analyzed using this method increases tremendously. Mass spectrometers are used in analytical systems in combination with a large range of ionization methods [3,4] (such as discharges, ions bombardment of surfaces in secondary ion mass spectrometry (SIMS), photoionization, electrospray ionization, and matrix-assisted laser desorption ionization (MALDI)). Therefore, nowadays, mass spectrometry is used intensively not only in chemistry and physics research [5] but also for applications in material [6] and environmental sciences [7], and in proteomics [8], metabolomics, or pharmaceutical fields [9].

Despite its utility in so many scientific fields, mass spectrometry also presents numerous limitations. One of the issues limiting the utilization of high-resolution instruments dedicated to specific applications (ICP-MS [10], SIMS, MS-GC, MALDI [9], etc.) is their high cost together with the necessity to use dedicated software packages.

In plasma physics practice, most of the mass spectra are recorded using easily available instruments, with limited mass resolution and accuracy. As a trade-off to the cost, in the spectra obtained using such instruments, the neighboring peaks cannot always be individually distinguished and confidently assigned to the mass species producing them. We could assume this issue is due to the fact that they may be displaced with respect to the expected position on the mass scale, superposed, and the data may be noisy. Moreover, in practice, the recorded data are expressed as a continuous sequence of isolated or superposed peak profiles, while the interpretation of the results is based on bar (also named centroid) representations [10] consisting of discontinuous sequences of separate bars whose heights indicate the concentrations of various mass species. The conversion of a profile mass spectrum to a bar mass spectrum may be a challenge, particularly in the case of physical systems consisting of many species with closed or equal values of their mass.

In this work, we present a procedure for processing profile mass spectra affected by inherent instrumental limitations such as peak superposition or mass position displacement, and their conversion to a bar representation. A novel analytical function is proved to be suitable for the description of the shape and asymmetry of the peak profile. The procedure is exemplified in the case of a magnetron plasma operated in an Ar and H_2_/Ar gas mixture with a W target, as the superposed spectra of W and WH ions provide good study material for the fitting procedure. The presented method may be further used for the study of plasma generated in hydrogen mixed with Ar, Ne, and N_2_ in contact with W materials, with direct applicability for the investigation of erosion and dust production caused by divertor sputtering in fusion technology [11]. 

The described procedure qualifies as a general guiding tool for the calibration of low-resolution mass spectrometers and the interpretation of mass spectra consisting of asymmetrical and wide peaks, being easily extendable to similar types of plasma sources. For example, peaks with similar shapes and large widths have previously been reported in the literature. Tyunkov et al. presented in [2], the ion mass-to-charge composition of the plasma generated by magnetron discharge with a titanium target and the study of the modification of the spectra when various gases are introduced in the discharge. The existence of five stable isotopes of Ti (with adjacent mass values) produces a superposed spectrum in which the area of each peak cannot be easily separated to obtain the concentration of each isotope.

To overcome the instrumental limitations and to extract the correct information, various strategies were adopted. Zhenga et al. [12] proposed a method for the peak detection and estimation of their half-width by using a continuous wavelet transform and then a curve fitting procedure. The reported procedure produces accurate results but is affected by the noise in the experimental data. Another method to analyze complex mass spectra recorded with limited mass resolution was presented by Stark et al. [13]. However, this method implies many steps prior to the fitting of the experimental data: a post-measurement *m*/*z* calibration, an algorithm to generate a list of all possible chemical formulas, determination of the number, position, and height of peaks, and later an iterative peak assignment algorithm. This kind of procedure offers good results and can be applied to complex organic systems, but the iterative fitting may encounter singularities, and the computation time is long. 

In the Experimental Section, we present the setup used for plasma generation by magnetron sputtering of a W target, and details on the mass spectrometer together with a short description of the recorded mass spectra and of the instrumental and physical aspects limiting the spectra interpretation. The next section, namely, Peak Fitting Procedure and Results, is divided into subsections, devoted to the criteria of selection of the fitting functions and to the procedures used for fitting individual mass peaks, for instrument calibration, and separation of isotopic contributions in the superposed peaks spectra region. The procedure is exemplified by determining the ratio between the tungsten hydride molecular ions and metallic tungsten ions in W/Ar/H_2_ plasma and by the conversion of profile mass spectra to more relevant bar mass spectra. Finally, in Conclusions, we point out the essential results of the procedure and the possibilities of extending it to other physical situations.

## 2. Setup Details and Description of the Experimental Mass Spectra

### 2.1. Experimental Setup and Operating Parameters

The schematic view of the experimental setup is presented in Figure 1a. Plasma is generated in a cylindrical chamber, in which a base pressure of 10^−3^ mbar is ensured by a combined roots and fore pumping system. A Torus magnetron (Lasker Co.), equipped with a 1-inch target, is powered by an RF generator (AD-TEC, AX series, 13.56 MHz) via an automated matching box (AD-TEC, AMV-1000-EN). A unit of gas admission and control made of two mass flow controllers (Bronkhorst) supplies the discharge with independent flows of Ar and H_2_. The two working gases are introduced into the discharge sideway, close to the magnetron target. The experiments reported here are carried out with an Ar or Ar/H_2_ mixture (1:1 ratio), using an RF power of 80 W, at a total pressure of 7.2 × 10^−2^ mbar.

Plasma sampling is realized by using an energy-dispersive quadrupole mass spectrometer (MS) (Hiden-Analytical EQP 1000, mass range 0–300 amu) equipped with an extraction aperture of 100 microns in diameter. This is placed in front of the magnetron head, at 6 cm distance. The MS is operated with the ionization source off, in the Positive Ions mode. The energy distributions of various ions formed in the W/H_2_/Ar plasma are monitored, and the count rates for the collected ions are recorded at an energy of ions set to 12 eV close to the maximum of the W ions energy distribution, with a minimum mass step of 0.01 amu.

For the instrumental calibration, mass spectra are acquired in Ar plasma with a dedicated multi-element (ME) target. A photo of the target is shown in Figure 1b. It consists of a 1-inch Al disk in which small cylinders of Cu, Ag, W, W–Th, and Au are incorporated, along the erosion track. These elements are selected for their capability to provide elemental mass peaks distributed over the entire analysis range (0–300) of the spectrometer.

### 2.2. Description of the Recorded Mass Spectra

A general mass spectrum obtained from the Ar/W plasma is presented in Figure 2a. The main species peaks correspond to the Ar ions and W ions. The spectrum is also polluted with contaminants’ peaks arising from the low vacuum and gas desorption from the walls. The most important contaminant peaks belong to H, C, N, and O species and their combinations. In particular, the previous use of acetylene for the study of carbon materials deposition is clearly seen by the presence of a H^+^ signal and the C_x_H_y_^+^ species. Even more, in the case of the mass spectrum obtained from the Ar/H_2_/W plasma (Figure 2b), the amplitude of molecular ionic species related to hydrogen presence is amplified, for example, in the zone of hydrogen (H^+^, H_2_^+^, and H_3_^+^), hydroxyl OH^+^, argon (ArH^+^), and W region (WH^+^).

The spectrum obtained from the ME target (Figure 2c) presents similar features, but is much richer in peaks. For example, the peaks corresponding to Cu^+^ (one dominant isotope, 62.93 amu), Ag^+^ (two isotopes at 106.905 and 108.904 amu), Au^+^ (one isotope at 196.996 amu), and Th^+^ (one isotope at 230.035 amu) are clearly separated. In addition, the presence of Ni, Cr, and Fe species is noticed, with their origin being attributed to the sputtering of the stainless-steel-made magnetron shield.

Small deviations of the peak position with respect to the expected value for the ratio between mass and charge (*m/z*) are noticed. These deviations are systematically determined, as shown in Table 1.

Another aspect that makes the interpretation difficult is the shape of the recorded peaks. Well-isolated peaks, corresponding to Ar^2+^, Ar^+^, and ArH^+^ ions, are shown in Figure 3a,b. Additional examples of the peak profiles can be seen in Appendix A section. It is worth mentioning that the peaks associated with the mentioned species have asymmetric and trapezoidal shapes. 

With respect to the W mass region (Figure 3c,d), the assignation of the peak positions is far more difficult [14,15]. The specific limitation to this process is the superposition of the peaks. W has five isotopes, whose mass values and natural abundances are presented in Table 2. The peak superposition combined with the shape profile and the inherent noise makes a simple interpretation of the spectrum almost impossible (see Figure 3c). An additional difficulty appears in the case of hydrogen contamination or hydrogen–tungsten plasmas due to the small signal affected by noise and possible contributions of newly formed molecular WH^+^ species (Figure 3d).

A procedure for peak profile fitting and spectra calibration is developed and used, as described below, for profile spectra processing, separation of different species contributions in superposed spectra, and conversion of results to more relevant bar spectra.

## 3. Peak Fitting Procedure and Results

### 3.1. Selection of the Peaks Fitting Function

The measured signal from ions should be, in theory, a delta function; however, in reality, the recorded signal profile contains contributions from both physical events and instrumental distortions. Thus, the resulting spectrum is the convolution (Equation (1)) of the ion signal with the spread function of the instrument [16]. In mass, optical, and X-ray spectroscopy, the recorded events are generally considered normally distributed [17]. The peak profiles, shown in Figure 3, present a specific trapezoidal shape, with an oblique plateau at the top that cannot be described by the Gauss, Lorentz, or Voigt functions that are usually used in spectroscopy. A reasonable choice for the description of the observed profile is a *model function* obtained from the convolution of a Gaussian and a top hat function. The origin of the top hat function comes from the ion detection phenomenon. This is, in turn, due to the spectrum modification by the detection slit that can be described by the convolution of the spectrum with a rectangular shape function [18]. Moreover, the asymmetry of the peak profile has its origin in the instrumental profile, and such features are normal in the spectra recorded with quadrupole mass spectrometers [19]. Other justifications for the different heights at the lateral edges of our spectra can be due to the non-paraxial ions reaching the detector [20]. For simplicity, we chose to consider only instrumental distortion and introduce the asymmetry in the top hat function. In order to comply with the shape of the observed peak profiles, we modified the classic top hat function to have an oblique plateau (Equation (2)) and convoluted it with a Gauss function (Equation (3)). The resulting function, described by Equation (4), is an appropriate choice to describe both the trapezoidal shape and the asymmetry of the peaks profile. We will name this profile the “peak model function”:(1)$$F\left(x\right)=\left(H*G\right)\left(x\right)=\int_{-\infty }^{\infty }G\left(x-\xi \right)H\left(\xi \right)\;d\xi$$
(2)$$H\left(x,W,\alpha ,c,r\right)=r\;\frac{W+\alpha \left(x-c\right)}{W}\;\left\{\begin{array}{c} 0\;\;if\;\;\;\;\;\;\;\;\;\;\;\;\;\;\;\;x-c<-\frac{W}{2}\\ 1\;\;if\;\;\;\;x-c\;\in \left(-\frac{W}{2},\frac{W}{2}\right)\\ 0\;\;if\;\;\;\;\;\;\;\;\;\;\;\;\;\;\;\;\;\;x-c>\frac{W}{2}\; \end{array}\right.$$
(3)$$G\left(x,w,c,r\right)=\frac{r}{w}\;\sqrt{\frac{6}{\pi }}\;e^{-\frac{6{\left(x-c\right)}^{2}}{w^{2}}}$$
(4)$$\begin{array}{ll} F\left(x,w,W,\alpha ,c,r\right) & \\ & =r\;[\frac{w\;\alpha }{\;\;2\;\sqrt{6\;\pi }}\;\left(-e^{-\frac{3\left(2\left(x-c\right)-W^{2}\right)}{w^{2}}}+e^{-\frac{3\left(2\left(x-c\right)+W^{2}\right)}{w^{2}}}\right)\\ & +\;\;\;\;\;\;\;\;\;\;\;\;\;\;\;\;+\frac{1+\left(x-c\right)\alpha \;W}{2W}\;(-\mathrm{erf}\left[\;\frac{\sqrt{\frac{3}{2}}\;\left(2\;\left(x-c\right)-W\right)}{w}\;\right]\\ & +\mathrm{erf}\left[\;\frac{\sqrt{\frac{3}{2}}\;\left(2\;\left(x-c\right)+W\right)}{w}\;\right]\;\;)\;\;\;] \end{array}$$

Further on, we will explain the parameters inserted in Equations (1)–(3). The shape of peak model function *F* depends on the Gaussian width *w* and the oblique hat parameters, namely, its width *W* and obliquity *α*. The center position is given by the *c* parameter, while the area of an individual peak is given by the scaling parameter *r*. The last-mentioned parameter, *r*, is relevant for the measurement due to its physical meaning, namely, it indicates the respective species concentration. 

The shape of the initial and resulting functions for a specific choice of parameters is presented in Appendix A. The total width of the convolution is larger than the widths of both functions and is given in Equation (5):(5)$$W_{tot}=\sqrt{W^{2}+w^{2}}$$

Having the *peak model function* given by Equation (4), one may construct a model for the experimental spectrum as the sum of *N* functions *F*, where *N* is the number of individual peaks in the spectrum. We will refer to this function as *spectrum model function S*(*x*) (see Equation (6)) and use it for the simultaneous fitting of more peaks:(6)$$S\left(x,w_{1},\;\ldots w_{N},\;W_{1},\ldots W_{N},\alpha_{1},\ldots \alpha_{N},c_{1},\ldots c_{N},r_{1},\ldots r_{N}\right)=\overset{N}{\underset{i}{\sum }}F_{i}\left(x,W_{i},w_{i},\alpha_{i},c_{i},r_{i}\right)$$

When the mass spectra contain more isotopes of the same element, the ratio between the area of each peak and the total area of the spectrum should be equal to the natural abundance (NA) of the respective isotope. When fitting the experimental data, NA can be imposed in the *spectrum model function S*. Therefore, we can construct the function *S* such that the ratio between the peak area and the total area of the spectrum is fixed to the NA. The constructed model spectrum is then scaled to the experimental data through a single free parameter, named area scaling coefficient and denoted *R*.

### 3.2. The Fitting Procedure and Its Validation on Individual Experimental Peaks

We processed the mass spectra with a *Python program* that uses the function “*curve fit*” [21]. This function employs the *non-linear least squares method* [22,23] to fit a set of data with the *spectrum model function S*, starting with a set of initial parameters chosen by the operator. The quality of the fit is estimated by computing the normalized root mean square error (*NRMSE*) [24] with the expression given in Equation (7).
(7)$$NRMSE\left(\%\right)=\frac{\sqrt{\frac{1}{N}\sum_{i}^{N}{\left(Y_{i}-Y_{i}^{fit}\right)}^{2}}}{\frac{1}{N}\;\sum_{i}^{N}Y_{i}\;}*100$$

The fitted spectra of Ar^2+^, Ar^+^, ArH^+^, Ag^+^, Au^+^, Cu^+^, and Th^+^ ions, previously shown in Figure 3 and Appendix A, are presented in Figure 4. The variable parameters in each fit are *c*, *w*, *W*, *α*, and *r*. The initial values for the fitting parameters are selected by visual appreciation. The fits are of good quality, as evidenced by the accurate description of the asymmetric shape of the recorded peaks, but also by the low NRMSE values (mentioned in each figure).

### 3.3. Spectra Calibration for Peak Position and Peak Widths

Applying the fitting procedure to the well-identified peaks spread over the whole mass scale of the ME spectrum offers the ground to calibrate the instrument in the mass scale (via parameter *c*) and to find the dependence of the peak shape parameters (widths *W*, *w,* and obliquity *α*) upon peak position. Indeed, the fitting provides well-defined values for the centers of the peaks (parameter *c*), which can be compared with the real physical positions where they should have been placed. Thus, the deviations noticed in Table 1 can be exactly calculated and a calibration curve (*m*/*z*)*_real_* = *f*(*m*/*z*)*_rec_* can be drawn. The curve will be useful to assign species to unidentified peaks when the recorded positions are displaced with respect to the right positions. The mass calibration curve, as obtained from the fitting of individual peaks selected from the ME target spectrum, is presented in Figure 5, and at the scale of the figure, it looks linear with slope 1. Still, deviations from the expected peak positions exist, as shown in the inset of the figure describing the dependence of the relative error on *m*/*z* value,$\;\frac{{\left(\frac{m}{z}\right)}_{real}-{\left(\frac{m}{z}\right)}_{rec}\;}{{\left(\frac{m}{z}\right)}_{real}\;}=f{\left(\frac{m}{z}\right)}_{real}$, where relative errors ranging up to 2% are noticed.

The variations in the peak shape parameters, i.e., the Gaussian *w*, oblique hat width *W*, and obliquity *α*, upon the (*m/z*)*_real_* value for the well-identified peaks are presented in Appendix A. For the determination of the parameter’s values in any other *m/z* positions, linear fits are found appropriate:(8)y=b+ax
where *y* represents the shape parameters, *W*, *w*, and *α*, while *x* is the real *m*/*z* value. The best values for the *a* and *b* constants are mentioned in each case in Figure 5 and Appendix A graph legends. The utility of the determined dependences of *c*, *w*, *W*, and *α* parameters upon the expected position in the *m*/*z* scale consists of the prediction of appropriate values for the parameters that are entered in the fitting procedure in the case of processing superposed peaks regions, as is exemplified below for W^+^ and WH^+^ mass spectra.

### 3.4. Validation of the Fitting Procedure for Superposed Peaks

The validation of the fitting procedure is tested by comparing the isotope abundances computed by processing the W peak region, similar to what is presented in Figure 3c, with the naturally existing abundances (NA). The computed abundance (CA) is obtained as the ratio between the area from one peak and the total area of the element area of the spectrum in the fitted region. The calculations considered only the ^182^W, ^183^W, ^184^W, and ^186^W isotopes; the contribution of the peak at ^180^W being considered negligible due to the small natural abundance of only 0.12%. 

First, the spectrum is corrected with respect to peak displacements, using the scale calibration relation (Equation (8), parameters in the legend of Figure 5). After scale correction, calculations were performed in two variants:(A)With five free-fitting parameters for each isotope (*c* for peak for position; *w, W,* and α for peak shape; and one fitting parameter *r_i_* for isotope concentration—represented as the area of the peak). The *spectrum model function S* has its form defined in Equation (6) and depends on 20 free parameters, 5 for each peak.(B)With the isotope concentrations (*r_i_*) as the only free parameters, the position and shape parameters for each peak are extracted from the previous calibration curves (relations are shown in the legend of Appendix A). The *spectrum model function S = S*(*x*, *r_1_*, *r_2_*, *r_3_*, *r_4_*) depends on four parameters, representing the areas of each peak.

The fitted spectra of W regions for the W and ME targets in Ar (one fit parameter per peak) are presented in Figure 6a,b, while the results of the fitting are presented in Table 2. A comparison between the results from the fitting with five and one free parameter per peak is presented in Appendix A.

Since the nonlinear least square method minimizes the square of the residuals [23], we have lower NRMSE (presented in Appendix A) values for the spectra fitted with all twenty free parameters. However, it is remarkable that the procedure provides acceptable abundance values with less than 5% average error per peak. The still-reasonable NRMSE values obtained in the case of the fit with fixed parameters allow us to consider that the calibration method can impose physical restrictions on the fitting of the spectrum without a strong effect on the quality of the fit.

The spectra from the magnetron discharge in the Ar/H_2_ gas mixture have at least eight peaks in the W region, i.e., from the isotopes of tungsten and their molecular combination with hydrogen (hydrides, WH). Some of the WH peaks are centered in almost the same positions as the metallic W peaks, which makes peak separations difficult. Therefore, we used the calibration parameters to fit this spectrum (Figure 7): the peaks are centered in the expected *m*/*z* position, and the widths and obliquity parameters are fixed using the calibration lines. Because there is no physical reason for some of the isotopes to react selectively with hydrogen, the isotopic composition of the WH^+^ species must respect the isotopic composition (natural abundances) of the W^+^ species. By considering the NA of W^+^ and WH^+^ as fixed values, the need of having the area of each peak as a fitting parameter is essentially removed. Only two variable parameters, *R* and *RH,* enter the *spectrum model function*, *S*(*x*, *R*, and *RH*), where these parameters scale the fitted curve to the envelope of W^+^ and WH^+^ recorded superposed peaks.

The ratio between the scaling coefficient for the WH species and the sum of both scaling coefficients,
(9)$$\beta \;\left(\%\right)=\frac{RH}{R+RH}*100$$
gives the percentage of hydride ions with respect with all W species. For the W/Ar/H_2_ spectrum, recorded in the conditions detailed in the experimental section, we obtained a ratio value of 16.58%. One can take this result as proof that the fitting procedure and the calibration method developed in this work can lead to quantitative information about the ion concentrations in plasma, despite the superposition of peaks.

### 3.5. Reduction in Profile Spectra to Bar Spectra

As mentioned previously, bar spectra are more relevant than profile spectra, because they directly provide the concentration of the species via the height of the bars positioned at each species position. The conversion of the profile spectra to bar spectra is easily performed using the fitting procedure because the areas under the peaks are given by the values of the *r* parameters. Some representative bar spectra, obtained from the profile spectra presented in Figure 2 and Figure 3c,d, are shown in Figure 8.

## 4. Conclusions

Wide peak profiles, position displacements in the *m*/*z* scale, and peaks superposition are serious obstacles in interpreting mass spectra obtained with general-purpose spectrometers characterized by limited resolution and accuracy. In the present paper, these aspects are discussed in the case of W/Ar/H_2_ plasmas. We show that such experimental drawbacks can be overcome by fitting procedures, and propose a fitting function capable of correctly describing the shape of the peaks present in the spectra, whatever the peak width and position. 

The accuracy of the fitting procedure was demonstrated through the good match between the model curve and experimental peak profile data, and illustrated for a large variety of isolated peaks in magnetron plasmas generated in Ar with a multielement (W, Cu, Ag, Au, and W–Th) target. The fitting procedure provides the right positions and shapes of the peaks, allowing the spectra calibration of those parameters with respect to the *m*/*z* scale. 

The problem of separating the different species contributions from the spectra consisting of superposed peaks was illustrated for the W ions region in Ar and Ar/H_2_ plasmas. Such plasma systems are highly relevant for the wide research community working in fusion technology. Herewith, in the *m*/*z* range of 180–188 amu, the peaks of the W^+^ isotopes are not well separated, and they are superposed with those of the WH^+^ species. In such cases, we demonstrated that the fitting of the recorded spectrum with a function constructed as the sum of separate peak model functions provides the correct natural abundance of the W isotopes and allows for the determination of the percentage of the plasma-formed hydride WH^+^ ions with respect to the atomic W^+^ ions. 

Finally, we successfully converted profile spectra into bar spectra, offering a representation where the concentration of species is directly indicated by the height of individual bars. This transformation was facilitated by the fitting procedure, as the peak area was directly obtained from the fit.

In conclusion, we presented a method for obtaining valuable information on ion species in plasmas, even in conditions of using mass spectrometers with low resolution and accuracy. The method is based on spectra calibration and the fitting of experimental data with adequate model functions. The model function proposed in this work consists of the convolution between a Gaussian and an oblique hat function, and is capable of correctly describing the shape of the peaks presented in various parts of the spectra. The proposed method is simple, uses low computing resources, and is easily applicable to a large variety of plasmas.

## Figures and Tables

**Figure 1 molecules-28-05664-f001:**
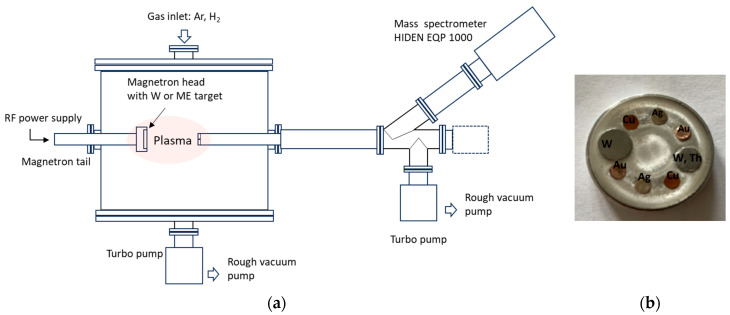
(**a**) Schematic view of the experimental setup and (**b**) image of the multielement target.

**Figure 2 molecules-28-05664-f002:**
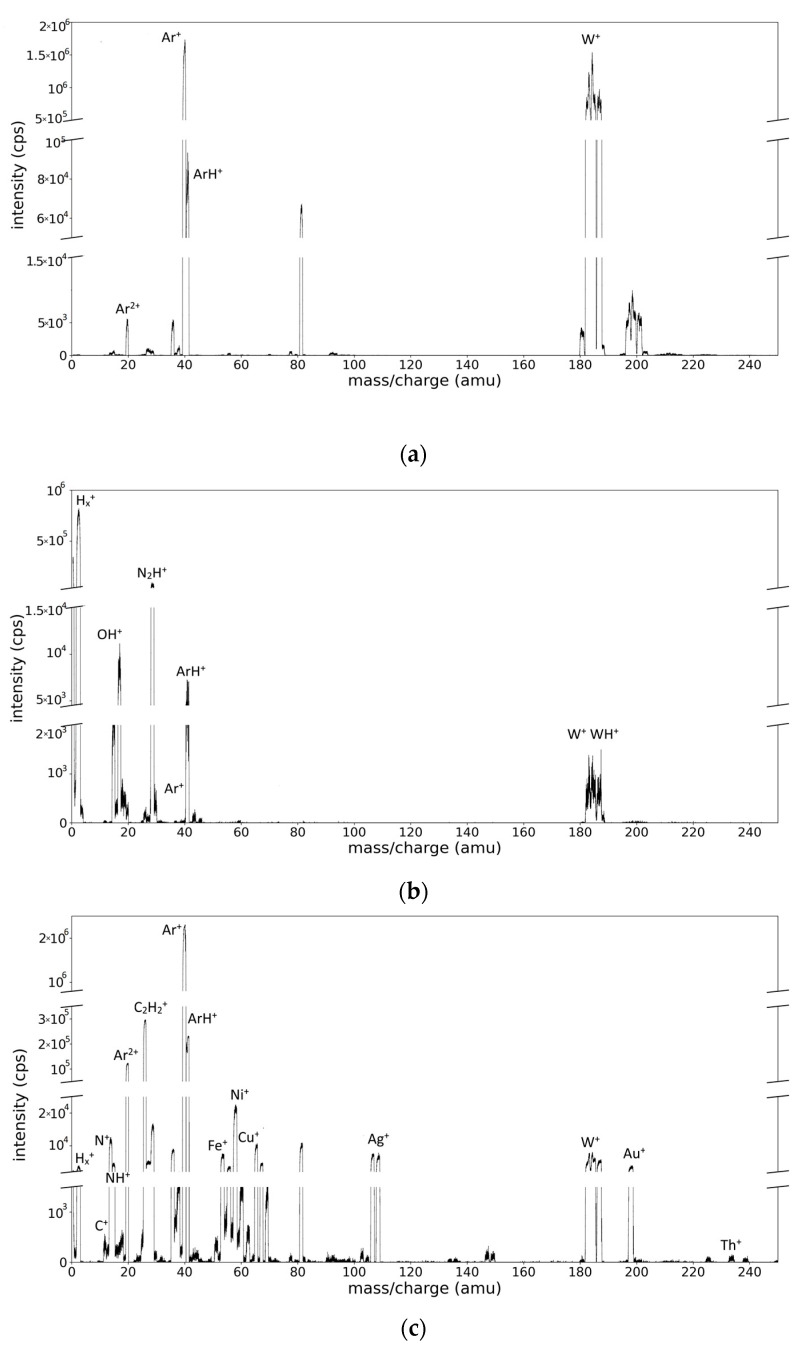
Mass spectra obtained: (**a**) from the plasma generated by the magnetron discharge with W target in Ar gas 40 sccm, p_c_ = 6.2 × 10^−2^ mbar and E_e_ = 17 eV (**b**) from the magnetron discharge with W target in Ar/H_2_ gas mixture 20/20 sccm, p_c_ = 6.5 × 10^−2^ mbar and E_e_ = 15.5 eV; (**c**) from the magnetron discharge with a ME target obtained in Ar (60 sccm) plasma at p_c_ = 4.5 × 10^−2^ mbar and E_e_ = 12.5 eV.

**Figure 3 molecules-28-05664-f003:**
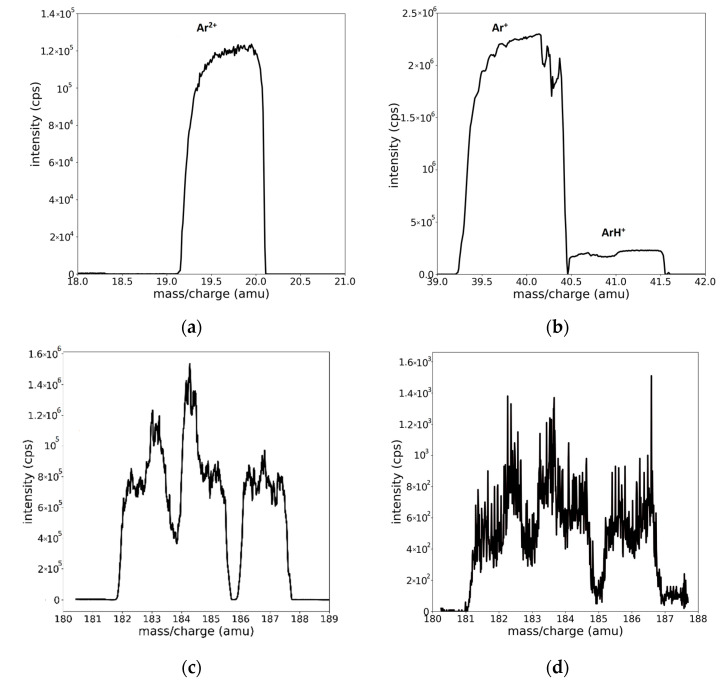
Well-isolated peak profiles presenting the asymmetric flat/trapezoidal shape of the (**a**) Ar^2+^, (**b**) Ar^+^, and ArH^+^ species. Mass spectra in the region of W, illustrating the superposition of isotope peaks, recorded for (**c**) W/Ar and (**d**) W/Ar/H_2_ plasmas.

**Figure 4 molecules-28-05664-f004:**
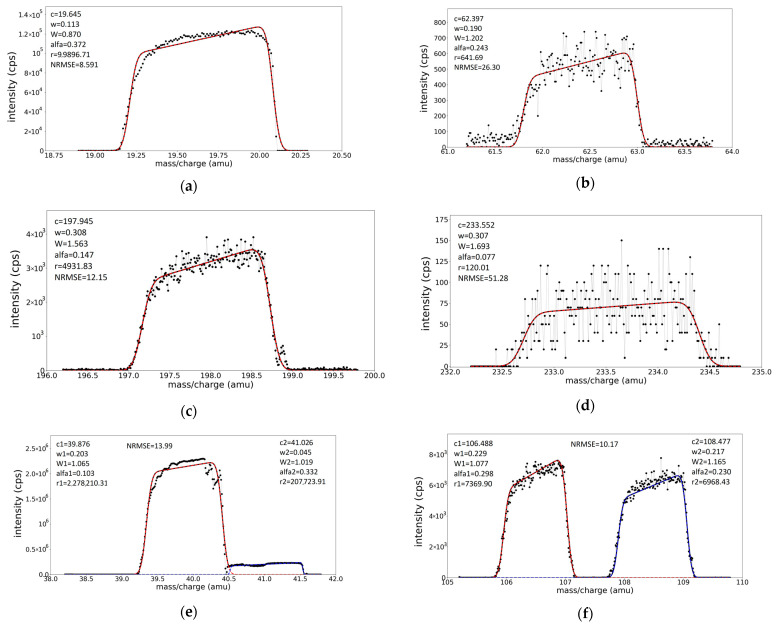
Fit (continous lines) of the well-isolated peaks in spectrum (recorded black points in the figures) on multielement target (**a**) Ar^2+^, (**b**) Cu^+^, (**c**) Au^+^, (**d**) Th^+^, (**e**) Ar^+^, ArH^+^, and (**f**) Ag^+^. The fitting procedure used for multielement spectra involved the varying of all five parameters for each peak present in the spectrum.

**Figure 5 molecules-28-05664-f005:**
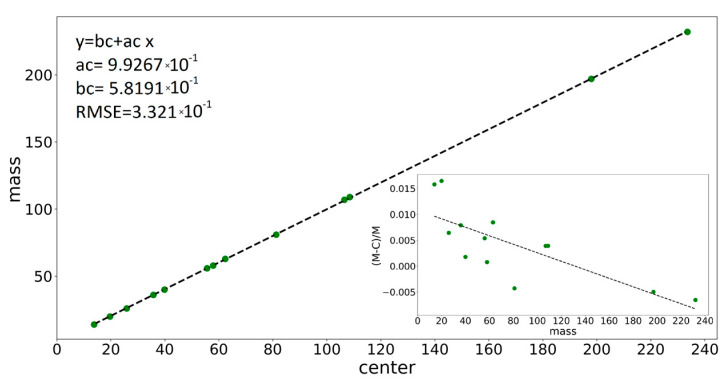
The variation in peak center with (*m*/*z*)*_real_* of the associated species, and in the inset, the dependence of measurement errors on (*m*/*z*)*_real_* value for various peaks in the ME spectrum together with the linear fit parameters and the RMSE value (the green points represent the experimentally measured mass values).

**Figure 6 molecules-28-05664-f006:**
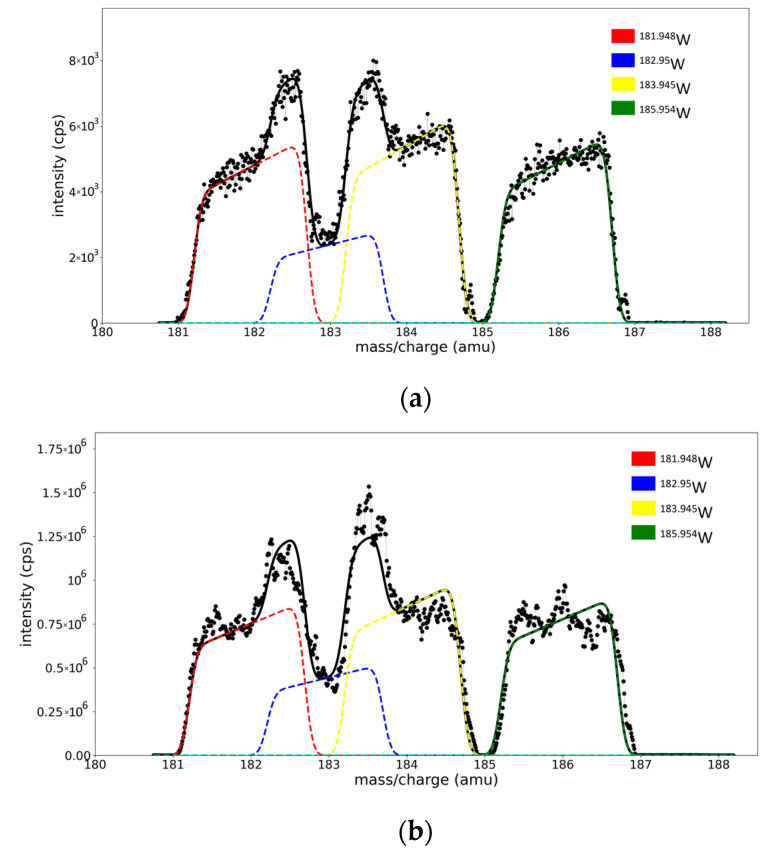
W region from (**a**) ME spectra and (**b**) W/Ar spectra. Black dots - the experimental data points, black line—the model spectra, colored line—separated peak for each isotope.

**Figure 7 molecules-28-05664-f007:**
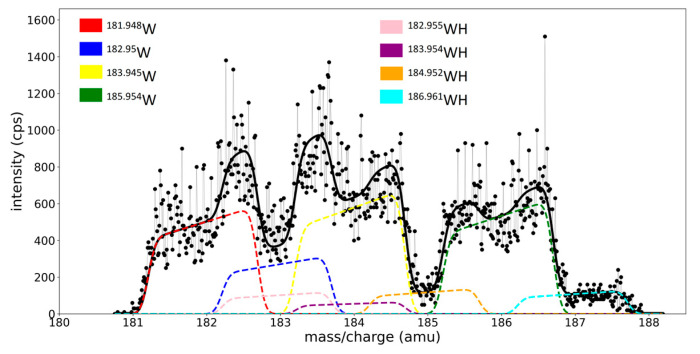
The fitted W region of the spectrum from the magnetron discharge with W target in Ar/H_2_ gas mixture.

**Figure 8 molecules-28-05664-f008:**
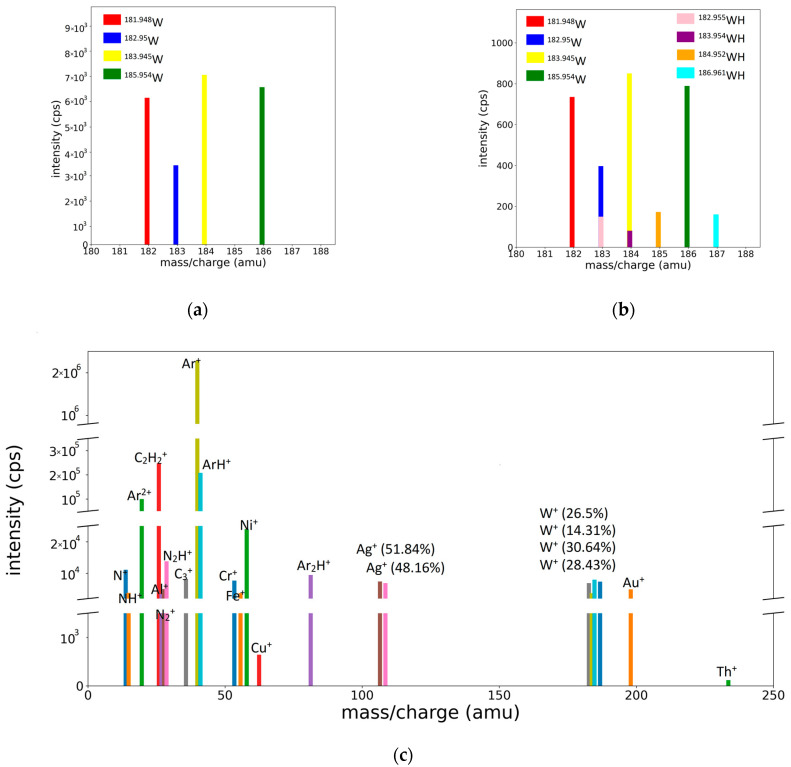
Bar representation of mass spectra: (**a**) for the W peaks of W/Ar, (**b**) W/Ar/H_2_ spectra, and (**c**) integral spectrum of the ME target.

**Table 1 molecules-28-05664-t001:** Deviation of the position of the recorded peaks with respect to the true *m*/*z* positions.

Ionic Species	Recorded Position (*m*/*z*)*_rec_*	Expected True Position (*m*/*z*)*_real_*	Deviation *D =* (*m*/*z*)*_real_ −* (*m*/*z*)*_rec_*	Relative Deviation *D_r_* (*%*) *= D*/(*m*/*z*)*_real_ × 100*
Ar^2+^	19.6	19.974	0.374	1.87
Ar^+^	39.8	39.948	0.148	0.37
ArH^+^	41.0	40.955	−0.045	−0.01
Cu^+^	62.4	62.93	0.53	0.84
^107^Ag^+^	106.5	106.905	0.405	0.37
^109^Ag^+^	108.4	108.904	0.506	0.46
^197^Au^+^	197.9	196.966	−0.934	−0.47
Th^+^	233.5	232.035	−1.465	−0.63

**Table 2 molecules-28-05664-t002:** The values for the natural abundance (NA) of W isotopes and the computed abundance (CA) obtained from the fit, together with the error calculated as the absolute value of the difference between NA and CA divided by NA and summed over all peaks.

Isotope	NA (%)	CA:ME (%)	$\frac{\left|NA-CA\right|}{NA}\times 100$	CA:W/Ar (%)	$\frac{\left|NA-CA\right|}{NA}\times 100$
^180^W	0.12	-		-	
^182^W	26.5	27.38	3.32	26.5	0
^183^W	14.31	13.66	4.54	15.73	9.94
^184^W	30.64	30.86	0.71	30.05	1.90
^186^W	28.43	28.08	1.23	27.70	2.56
$\sigma \left(\%\right)=\frac{1}{4}\sum \frac{\left|NA-CA\right|}{NA}\times 100$	-	-	2.45 ± 0.06		3.6 ± 0.11

## Data Availability

The data presented in this study are available on request from the corresponding author.

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
