# Peer review of "Mass Spectra Fitting as Diagnostic Tool for Magnetron Plasmas Generated in Ar and Ar/H2 Gases with Tungsten Targets"

_molecules, 2023, doi:10.3390/molecules28155664_

Round 1

Reviewer 1 Report

This study presents a method of processing mass spectra of ions from plasmas generated in Ar and Ar/H2 gases in contact with tungsten surface. The paper has abundant data, but there are still many problems to be improved. It is suggested to accept after modification.

Detailed review comments are as follows:

1) Please add some content in the abstract to describle the significant of this work.

2) The pictures have poor font and resolution, and can not see clearly. Please correct them. In addtion, please decrease the number of figers/tables.

3) In figure 6, why the mass/charge peak is so wide (about 1 Da width)? There should be a simple outline or bar diagram.

4)The content in conclusion is too much, please simplify it.

5) There are many inconsistencies in the references, especially in journal and other information, please modify the consistency.

6) Please cite the literature Mass spectrometry distinguishing C=C location and cis/trans isomers: A strategy initiated by water radical cations. Analytica Chimica Acta, 2020, 1139, 146-154 when describing a large range of ionization methods in introduction.

English language need improved

Author Response

Bucharest, July 17th, 2023

Dear Reviewer,

I am pleased to submit the documents related to the revised version of the manuscript titled "Mass Spectra Fitting as a Diagnostic Tool for Magnetron Plasmas Generated in Ar and Ar/H2 Gases with Tungsten Targets" for consideration and publication in Molecules journal.

The revised version is based on the initial submitted manuscript in which we performed modifications according to the Reviewers comments. The documents accompanying the submission are:

- Answers to Reviewer;

- Revised manuscript and Figures;

- Supplementary material file;

In the Answer to Reviewer file we considered in detail, point to point, the Reviewer comments, and recommendations. Beside answering to the raised questions, we presented in these documents the modifications that we performed in the manuscript, the most important being:

-We completed the Abstract section to emphasize the importance of the work;

-Section Introduction was enlarged, now including 4 new References, dedicated to previous works related to the subject;

-We re-edited the figures to increase the resolution and readability (please note that this resolution is partially lost during the conversion of the documents to the present .pdf file)  ;

-We created a new file, Supplementary material, where we moved some of figures presented initially in the original manuscript, and where we included also additional results;

-Section Conclusions was reformulated to be shorter and more conclusive.

We are confident that the present revised version of our work represents a significant improvement of the initial manuscript, is clearer and more fluent, and fulfils the high standards of publication in the Molecules journal.

With the best regards,

Gheorghe DINESCU

(on behalf of all co-authors)

Reviewer 2 Report

The authors present a work devoted to the processing mass spectra of ions from plasmas generated in Ar and Ar/H2 in contact with tungsten surfaces. Detailed mass spectra and pertaining profiles, along with peak fitting procedure, results and related validation, are presented and discussed in detail.

The work yields interesting results from both a fundamental and an applied point of view. I suggest the following modifications in order to improve the paper clarity, quality and impact:

- The front matter of the paper (Abstract, Introduction, Conclusions) should better emphasize the novelty and importance of this work with respect to the state-of-the-art in the field.

- The choice of the targeted case study must be motivated in detail.

- Figs. 1 and 2 can be merged in a single multi-panel one. The actual Fig. 1 is too schematic and must be significantly enriched with instrumental details.

- The graphical quality of figures must be significantly improved. This is particularly needed in the case of Figs. 3, 4, 6, 7. The font sizes must also be enlarged to improve the figure readability.

- Fig. 3: please remove experimental parameters from the figure and indicate them in the caption.

- Fig. 5 can be transferred into the Supporting Information section. The same holds for Table 3.

- Please carefully check the consistency of the significant figures for numerical values in the whole paper text and tables, and report also the corresponding uncertainties.

- Are the linear fittings in Figs. 7b,c,d really meaningful? Please motivate and elaborate supported observations in detail. In addition, experimental uncertainty bars on the reported data points must be marked in the figures.

- Lines 276: higher precision, or higher accuracy?

- Lines 305-306: please rephrase the sentence to clarify it.

- The Conclusions must be made shorter and less technical, and also describe the most attractive perspectives for further developments of the present research activities.

- A revision of the whole paper text with the aid of a native English speaker is recommended to polish the language and make it more fluent.

Author Response

Bucharest, July 17th, 2023

Dear Reviewer,

I am pleased to submit the documents related to the revised version of the manuscript titled "Mass Spectra Fitting as a Diagnostic Tool for Magnetron Plasmas Generated in Ar and Ar/H2 Gases with Tungsten Targets" for consideration and publication in Molecules journal.

The revised version is based on the initial submitted manuscript in which we performed modifications according to the Reviewers comments. The documents accompanying the submission are:

- Answers to Reviewer;

- Revised manuscript and Figures;

- Supplementary material file;

In the Answer to Reviewer file we considered in detail, point to point, the Reviewer comments, and recommendations. Beside answering to the raised questions, we presented in these documents the modifications that we performed in the manuscript, the most important being:

-We completed the Abstract section to emphasize the importance of the work;

-Section Introduction was enlarged, now including 4 new References, dedicated to previous works related to the subject;

-We re-edited the figures to increase the resolution and readability (please note that this resolution is partially lost during the conversion of the documents to the present .pdf file);

-We created a new file, Supplementary material, where we moved some of figures presented initially in the original manuscript, and where we included also additional results;

-Section Conclusions was reformulated to be shorter and more conclusive.

We are confident that the present revised version of our work represents a significant improvement of the initial manuscript, is clearer and more fluent, and fulfils the high standards of publication in the Molecules journal.

With the best regards,

Gheorghe DINESCU

(on behalf of all co-authors)

Round 2

Reviewer 2 Report

The revised paper will be acceptable after a careful language

revision with the aid of a native English speaker.

See above

Author Response

Dear Reviewer,
I am pleased to submit the documents related to the English proofed version of the manuscript titled "Mass Spectra Fitting as a Diagnostic Tool for Magnetron Plasmas Generated in Ar and Ar/H2 Gases with Tungsten Targets" accepted for publication in Molecules journal.
The English revised version is based on the accepted manuscript which was checked for English language by a proficient English user. It contains:
- Revised manuscript file where the modifications are marked with Track Changes;
- Supplementary material file, in which the minor English corrections were carried out directly in the text.
Thank you for the positive attitude and the good collaboration.
Best regards,
Gheorghe DINESCU
(on behalf of all co-authors)
